



# Improved representations of coupled soil-canopy processes in the CABLE land surface model

Vanessa Haverd[1], Matthias Cuntz[2], Lars P. Nieradzik[1], Ian N. Harman[1]

[1] CSIRO Oceans and Atmosphere, P.O. Box 3023, Canberra ACT 2601, Australia.

[2] Department Computational Hydrosystems, UFZ—Helmholtz Centre for Environmental Research, Permoserstr. 15, 04318 Leipzig, Germany

*Correspondence to*: Vanessa Haverd (Vanessa.haverd@csiro.au)

**Abstract.** CABLE is a global land surface model, which has been used extensively in offline and coupled simulations. While CABLE performs well in comparison with other land surface models, results are impacted by decoupling of transpiration and photosynthesis fluxes under drying soil conditions, often leading to implausibly high water use efficiencies. Here we present a solution to this problem, ensuring that modeled transpiration is always consistent with modeled photosynthesis, while introducing a parsimonious single-parameter drought response function which is coupled to root water uptake. We further improve CABLE's simulation of coupled soil-canopy processes by introducing an alternative hydrology model with a physically accurate representation of coupled energy and water fluxes at the soil/air interface, including a more realistic formulation of transfer under atmospherically stable conditions within the canopy and in the presence of leaf litter. The effects of these model developments are assessed using data from 18 stations from the global Eddy covariance flux network FLUXNET, selected to span a large climatic range. Marked improvements are demonstrated, with root-mean-squared errors for monthly latent heat fluxes and water use efficiencies being reduced by 40%. Results highlight the important roles of deep soil moisture in mediating drought response and litter in dampening soil evaporation.



# 1 Introduction

In many global terrestrial carbon-cycle models, global gross primary production (GPP) and net biome production (NBP) are over-sensitive to precipitation anomalies. This was reported by Piao et al. (2013) and highlighted in the IPCC 5[th] Assessment Report (Ciais et al., 2013): "Terrestrial carbon cycle models used in AR5 generally underestimate GPP in the

water limited regions, implying that these models do not correctly simulate soil moisture conditions, or that they are too sensitive to changes in soil moisture (Jung et al., 2007). Most models […] estimated that the interannual precipitation sensitivity of the global land $CO_2$ sink to be higher than that of the observed residual land sink (–0.01 PgC yr$^{-1}$ mm$^{-1}$; […])."

CABLE is the land surface scheme in the ACCESS earth system model (Kowalczyk et al., 2013; Law et al., 2015), as

used in the IPCC 5[th] Assessment report (Ciais et al., 2013), and is one of an ensemble of ecosystem and land-surface models contributing to the Global Carbon Project's TRENDY initiative (Ahlström et al., 2015; Sitch et al., 2015). While CABLE 2.0 performs well in comparison with other land surface models (e.g. Best et al., 2015), results suggest an over-sensitivity of evapotranspiration to drought (Best et al., 2015), and may be impacted by decoupling of transpiration and photosynthesis fluxes under drying soil conditions (Wang et al., 2011), potentially leading to implausibly high water use

efficiencies.

The responses of gross primary production (GPP) and evapotranspiration (ET) to water availability in CABLE have featured in recent studies by Li et al. (2012) and De Kauwe et al. (2015). Both studies noted an over-sensitivity of ET to water-availability in CABLE with the standard drought response setting. Li et al. (2012) implemented an alternate stomatal drought response function based on the parameterization of Lai and Katul (2000), along with a parameterization

for hydraulic redistribution (Ryel et al., 2002) and demonstrated marked improvements at three FLUXNET sites, largely attributable to the introduction of hydraulic redistribution.

De Kauwe et al. (2015) applied alternative drought responses to stomatal conductance and photosynthetic capacity, based on the formulations of Zhou et al. (2013). Improvements were demonstrated at five European FLUXNET sites, with model performance dependent on a site-specific drought tolerance parameter.

Haverd et al. (2013) proposed an alternative formulation for coupled drought response and root water extraction in CABLE, operating in tandem with an alternative soil hydrology scheme called SLI (Haverd and Cuntz, 2010). In that work, CABLE, constrained by multiple observation types, was applied to a high-resolution (0.05° x 0.05°) assessment of the Australian terrestrial carbon and water cycles. Here the constrained model, including an alternative drought response, performed well against Eddy-covariance based flux estimates, and in particular replicated the observed sustained

evapotranspiration through seasonal drought periods in drought-adapted savanna ecosystems.

In this work, we resolve in CABLE 2.0 the problems of over-sensitivity of ET to drought and decoupling of transpiration and photosynthesis fluxes under drying soil conditions. Firstly, we introduce the alternative drought-response of Haverd et al. (2013) as an option in CABLE2.0, making use of global data on maximum vegetation rooting depth, and ensuring that



photosynthesis is limited by extractible soil moisture. Since a significant component of ET can be soil evaporation, we secondly improve the physical accuracy of the modeled soil evaporation by accounting for the potentially significant effect of leaf litter on soil evaporation. Thirdly, we introduce the SLI hydrology scheme, which is in part characterized by coupled heat and moisture fluxes within the soil column and at the soil-air interface, and newly accounts for local stability

effects on the resistance of transfer from the ground to the canopy air-space. We assess the impacts of the three stages of developments on model performance, using 95 site-years of observation-based estimates of ET, sensible heat H, GPP, and WUE from 18 globally-distributed Eddy covariance flux sites.

## 2 Model Description

The CABLE global land surface model is documented by Wang et al. (2011)(CABLE1.4b) and Kowalczyk et al. (2013) (CABLE1.8). Briefly, CABLE consists of five components: (1) the radiation module describes radiation transfer and absorption by sunlit and shaded leaves; (2) the canopy micrometeorology module describes the surface roughness length, zero-plane displacement height, and aerodynamic conductance from the reference height to the air within canopy or to the soil surface; (3) the canopy module includes the coupled energy balance, transpiration, stomatal conductance and

photosynthesis of sunlit and shaded leaves; (4) the soil module describes heat and water fluxes within soil (6 vertical layers) and snow (up to 3 vertical layers) and at their respective surfaces; and (5) the ecosystem carbon module accounts for the respiration of stem, root and soil organic carbon decomposition. CABLE2.0 includes full biogeochemistry available via the CASA-CNP module (Wang et al., 2010), and differs otherwise from CABLE1.8 only by small bug fixes and by changes to the vegetation optical properties, as described by Lorenz et al. (2014). CABLE has been benchmarked

off-line (e.g. Best et al., 2015; Zhang et al., 2013; Zhou et al., 2012) and in coupled environments (Kowalczyk et al., 2013).

### 2.1 Drought Response and Root Water Extraction in CABLE2.0

#### 2.1.1 Standard Model Parameterisation

**Drought Response**

Canopy photosynthesis and transpiration are coupled via stomatal conductance, modeled for each of sunlit and shaded leaves as:

$$G_s = f_{w,soil}\left(G_0 + \frac{a_1 A_c}{\left(C_s - \Gamma^*\right)\left(1 + D_s/D_0\right)}\right) \qquad (1)$$





where $G_0$ is residual conductance [mol m$^{-2}$ s$^{-1}$], $D_s$, $C_s$ and $A_c$ are the water vapour pressure deficit at the leaf surface, $CO_2$ concentration at the leaf surface and net photosynthesis respectively; $\Gamma^*$ is the $CO_2$ compensation point of photosynthesis in the absence of mitochondrial respiration other than that related to photorespiration [mol m$^{-1}$] (a function of canopy temperature), $a_1$ and $D_0$ are two model parameters, and $f_{w,soil}$ is the stomatal conductance drought response factor, calculated as:

$$f_{w,soil} = \beta_v \sum_j g_j \frac{\theta_j - \theta_w}{\theta_{fc} - \theta_w} \qquad (2)$$

where $\beta_v$ is a model parameter, $g_j$ is the fraction of root mass in the $j^{th}$ layer, $\theta_j$ is the volumetric soil moisture content of the $j^{th}$ soil layer, $\theta_w$ and $\theta_{fc}$ are volumetric soil water contents at wilting point and field capacity respectively.

In CABLE, 6 vertical soil layers (thicknesses from the top to bottom: 2.2 cm, 5.8 cm, 15.4 cm, 40.9 cm, 108.5 cm, 287.2 cm) are represented, with soil moisture and temperature state variables updated using one-dimensional Richard's and energy continuity equations respectively. The cumulative root density distribution function and associated plant-functional-type (PFT) specific parameter $\beta$ of Jackson et al. (1996) is adopted:

$$\sum_{j=1}^{k} g_j = 1 - \beta^{z_k} \qquad (3)$$

where $z_k$ is the depth to the bottom of the $k^{th}$ layer.

**Coupled Transpiration and Photosynthesis**

Coupled equations for net photosynthesis and energy balance (Wang and Leuning, 1998) are solved iteratively, providing an initial solution for the transpiration flux, $q_{trans,0}$ [m s$^{-1}$] that is consistent with the stomatal conductance and net photosynthesis.

**Actual Transpiration**

This value of transpiration may then be adjusted down according to soil water availability, giving an actual transpiration flux:

$$q_{trans} = \sum_j \min\left[ q_{trans} g_j \Delta t, \max\left[ 0.0, \left( \theta_j - 1.1\theta_w \right) \Delta z_j \right] \right] \qquad (4)$$

In Equation (4), $\Delta t$ is the model time step [s] and $\Delta z_j$ [m] is the thickness of the $j^{th}$ soil layer. The surface energy balance is calculated with this adjusted value of transpiration, but net photosynthesis is not, which leads to a decoupling of carbon and water fluxes whenever the demand for root water extraction exceeds availability.

**Root Water Extraction**

Demand for root water extraction in the $j^{th}$ layer is set to $q_{trans} g_j \Delta t$, where $q_{trans}$ is the transpiration rate [m s$^{-1}$]. Actual root extraction in each layer, $r_{ex,j}$ [m s$^{-1}$] is the lesser of the extractible water and the demand for root water extraction augmented by the demand from layers above that are also in excess of extractible water:



$$r_{ex,j} = \frac{1}{\Delta t} \min\left\{ \left(\theta_j - \theta_w\right)\Delta z_j, \, g_j q_{trans}\Delta t + \sum_{k=1}^{j-1}\max\left[0.0, g_k q_{trans}\Delta t - \left(\theta_k - \theta_w\right)\Delta z_k\right]\right\} \tag{5}$$

### 2.1.2 Modified Model

**Coupled drought response and root water extraction**

The rate of root-water uptake from level $j$ is modelled as:

$$r_{ex,j} = \alpha(\theta_j) g_j q_{trans} \tag{6}$$

where $g_j$ is the fraction of fine root mass in the $j^{th}$ layer and $q_{trans}$ is the actual transpiration rate [m s$^{-1}$], here equal to the transpiration rate $q_{trans,0}$ that is determined from the coupled equations for leaf energy balance and net photosynthesis. $\theta_j$ is the volumetric liquid soil moisture content, and $\alpha(\theta)$ is proportional to the root "shut-down" function of Lai and Katul

(2000):

$$\alpha_1(\theta) = \begin{cases} \left(\dfrac{\theta - \theta_w}{\theta_s}\right)^{\gamma/(\theta-\theta_w)} & \left(\theta - \theta_w\right) > 0 \\ 0 & \left(\theta - \theta_w\right) \leq 0 \end{cases} \tag{7}$$

where $g$ is an empirical parameter controlling the rate at which $\alpha_1(\theta)$ approaches 0. $\alpha(\theta)$ is rescaled from $\alpha_1(\theta)$ such that $\sum r_{ex,j} = q_{trans}$ :

$$\alpha_j = \frac{\alpha_1(\theta_j)}{\sum\limits_{k}\alpha_1(\theta_k) g_k} \tag{8}$$

We then test for over-extraction in each of the $j$ layers separately, and scale $\alpha_j$ by a factor $\left(\theta_j - \theta_w\right)\Delta z_j / \left(1.1 q_{trans} dt\right)$ if the current value of $\alpha_j$ will reduce soil moisture below the wilting point. If a re-test still yields over-extraction, we force total extraction to zero by setting $f_{w,soil} = 0$.

Otherwise, the stomatal drought response depends on the soil moisture content of the wettest accessible layer:





$$f_{w,soil} = \max\left\{\alpha_1(\theta_j)\delta_j, j = 1, n\right\} \tag{9}$$

where $\delta_j = 1$ when the upper layer bound is less than a PFT-dependent maximum rooting depth ($z_r$) $\delta_j = 0$, and $n$ is the total number of soil layers. Equation (9) is an attempt to capture the ecological optimality hypothesis that evolutionary selection pressures drive ecosystems towards maximal utilization of available resources (Raupach, 2005), without imposing an optimal carbon allocation scheme. Maximum rooting depths (Table 1) are set according to the depth at which the cumulative root fraction from the surface is 99%, as estimated by Zeng (2001), using data from Canadell et al. (1996).

**Table 1: CABLE parameter values for maximum rooting depth ($z_r$) and above-ground fine structural litter ($C_{litt}$)**

| PFT | $z_r$ (m) | $C_{litt}$ (tC ha$^{-1}$) |
|---|---|---|
| Evergreen needleleaf forest | 1.8 | 20.0 |
| Evergreen broadleaf forest | 3.0 | 6.0 |
| Deciduous needleleaf forest | 2.0 | 10.0 |
| Deciduous broadleaf forest | 2.0 | 13.0 |
| Shrub | 2.5 | 2.0 |
| C3 grassland | 1.5 | 2.0 |
| C4 grassland | 2.4 | 0.3 |
| Tundra | 0.5 | 0.3 |
| C3 cropland | 1.5 | 0.0 |
| C4 cropland | 1.5 | 0.0 |
| wetland | 1.8 | 2.0 |



Equations (6)-(9) are evaluated after each call to the subroutine that solves the coupled equations for stomatal conductance, photosynthesis and leaf energy balance, which includes the calculation of the transpiration rate. Since this subroutine is called 4 times within a loop in which atmospheric stability is iteratively updated, updates to $f_{w,soil}$ feed back to coupled transpiration and photosynthesis. In the extreme case where the initial transpiration estimate leads to $f_{w,soil} = 0$,

the subsequently calculated transpiration and photosynthesis are zero, and all net radiation absorbed by the leaf is converted to sensible heat. This is in contrast to the default model where photosynthesis may proceed in the absence of extractible water.

## 2.2 Soil surface energy balance

### 2.2.1 Standard model

The latent heat flux, $\lambda E_{soil}$ [W m$^{-2}$], and sensible heat flux, $H_{soil}$ [W m$^{-2}$] from the soil are calculated as follows:

$$\lambda E_{soil} = \min\left[ c_w \lambda \Delta z_1 \left(\theta_1 - \theta_w\right)/\Delta t, w_s\left( \Gamma\left(R_{net,soil} - G_0\right) + \left(1 - \Gamma\right)\frac{\lambda \rho_a\left(q^*(T_{soil,1}) - q_c\right)}{r_{soil}} \right) \right] \quad (10)$$

$$H_{soil} = c_p \rho_a \left(T_{soil,1} - T_a\right)/r_{soil} \quad (11)$$

The latent heat flux at the soil surface is the lesser of a supply and demand term, where the demand term is calculated as the Penman-Monteith potential evaporation, scaled down by a soil wetness factor. In Equations (10) and (11), $c_w$ is the

density of water [kg m$^{-3}$], $\Delta z_1$ is the thickness of the top soil layer [m], $w_s$ is a soil wetness factor, $\lambda$ the latent heat of fusion [J kg$^{-1}$], $\rho_a$ the density of air [kg m$^{-3}$], $\Gamma = s/\left(s+\gamma\right)$, $s$ is the slope of saturated vapour pressure with respect to temperature [m$^3$(H$_2$O) m$^{-3}$(air) K$^{-1}$], $c_p$ the heat capacity of dry air [kg m$^{-3}$ K$^{-1}$], $\gamma = c_p/\lambda$ is the psychrometric constant, $q^*$ is the saturated specific humidity [kg kg$^{-1}$], $q_c$ is in-canopy specific humidity [kg kg$^{-1}$], and $r_{soil}$ is the resistance to turbulent transfer from the soil/air interface to the displacement height [s m$^{-1}$]. The soil wetness factor scales

down the Penman-Monteith potential evaporation, and is calculated as:

$$w_s = \min\left[ 1, \frac{\theta_1 - 0.5\theta_w}{\theta_{fc} - 0.5\theta_w} \right]. \quad (12)$$

Net radiation absorbed by the soil ($R_{net,soil}$) is calculated as the sum of shortwave and longwave components (Wang et al., 2011), where the longwave component depends on the surface soil temperature (assumed the temperature of the top soil layer) from the previous time step. The ground heat flux ($G_0$) is calculated as the residual of the surface energy balance

from the previous time-step.





The resistance $r_{soil}$ is formulated as the integral over height $z$ of the inverse Eddy diffusivity from the roughness length of the soil ($z_{0s}$) to the displacement height in the canopy ($d$):

$$r_{soil} = \int_{z_{0s}}^{d} \frac{dz}{\sigma_w^2 \tau_L} \qquad (13)$$

where the vertical velocity standard deviation is formulated as:

$$\sigma_w = u_* a_3 \exp\left\{ c_{sw} L \left( \frac{z}{h} - 1 \right) \right\} \qquad (14)$$

and the Lagrangian time-scale as:

$$\tau_L = \left( \frac{c_{TL} h}{u_*} \right) \frac{z}{d} \qquad (15)$$

where $a_3$ and $c_{TL}$ are constants with respective values of 1.25 and 0.40; $L$ is leaf area index; $u_*$ the friction velocity at the top of the canopy; $h$ the canopy height.

The default model uses an approximation to the integral in Equation (13):

$$r_{soil} = \ln\left\{ \frac{d}{z_{0S}} \right\} \frac{\exp\{2c_{sw} L\} - \exp\left\{ 2c_{sw} L \left( 1 - \frac{d}{h} \right) \right\}}{a_3^2 c_{TL} 2 c_{sw} L} \qquad (16)$$

as used by Raupach et al. (1997) and subsequently propagated to CABLE (Wang et al., 2011, Eq A.14). However the analytic form of the integral is (Haverd et al., 2013):

$$r_{soil} = \frac{1}{u_*} \ln\left\{ \frac{d}{z_{0s}} \right\} \frac{\exp\{2c_{s,w} L\} \left( \frac{d}{h} \right)}{a_3^2 c_{TL}} \qquad (17)$$

and results in higher values of $r_{soil}$.

### 2.2.2 Leaf litter effects on surface energy balance

Resistances to heat and water vapour transfer at the soil/air interface are augmented by a component representing the effect of litter:

$$r_{bh} = r_{soil} + \frac{\Delta z_{litt}}{\rho_a k_{H,litt}} \qquad (18)$$



$$r_{bw} = r_{soil} + \frac{\Delta z_{litt}}{D_{v,litt}} \tag{19}$$

where $\Delta z_{litt}$ is the depth of fine structural litter[m], $k_{H,litt}$ is the thermal conductivity of the litter layer. The depth of the litter layer is

$$\Delta z_{litt} = \frac{2.0 C_{litt}}{\rho_{litt}} \tag{20}$$

where $C_{litt}$ is the above-ground fine structural litter pool [kg(C) m$^{-2}$], inherited here on a PFT-basis from the carbon-cycle component of the model, under the assumption that half the total fine structural litter (derived from leaf and root turnover) is stored above-ground. Values of $C_{litt}$ are given in Table 1. These were obtained by running the model with biogeochemistry enabled (carbon-cycle only: nitrogen- and phosphorous-cycles were disabled) using repeated GSWP-2 3-hourly meteorology for the 1986-1995 period (Dirmeyer et al., 2006) until carbon pool convergence was achieved.

The factor of 2.0 in Equation (20) converts from mass of carbon to mass of dry matter, and $\rho_{litt}$ is the bulk density of litter, here 62 kg m$^{-3}$ (Matthews, 2005). Vapour diffusivity within the litter is estimated using the empirical formulation of Matthews (2005):

$$D_T(z_{Litt}) = D_{T0} \exp\left\{ \chi\left( \frac{z_{Litt}}{\Delta z_{Litt}} - 1 \right) \right\} \tag{21}$$

$$D_{T0} = D_{T0,a} \exp\left( U D_{T0,b} \right) \tag{22}$$

$$\chi = \chi_a + U \chi_b \tag{23}$$

where $z_{Litt}$ is the depth within the litter (set here to 0.5 $\Delta z_{Litt}$); $U$ is windspeed 10 cm above the litter surface and $\chi_a$, $\chi_b$, $D_{T0,a}$ and $D_{T0,b}$ are empirical coefficients with respective values of 2.08, 2.38 m s$^{-1}$, 2·10$^{-5}$ m$^2$ s$^{-1}$, and 2.60 m$^{-1}$ s.

Heat conductivity of the litter layer is also taken from Matthews (2005):

$$k_{H,L} = 0.2 + 0.14 \theta_{Litt} \frac{\rho_w}{\rho_{Litt}} \tag{24}$$

Here $\theta_{Litt}$ is the volumetric moisture content of the litter. For reasons of computational efficiency, and unlike Haverd and Cuntz (2010), we do not solve for $\theta_{Litt}$, instead assuming a fixed value of half of the saturated moisture content, here taken as 0.09 (Matthews, 2005)





### 2.2.3 SLI soil model

**Surface Energy Balance**

The SLI (Soil-Litter-Iso) model (Haverd and Cuntz, 2010; Haverd et al., 2013) extends Ross' fast numerical solution (Ross, 2003) of Richard's Equation to include coupled vertical heat and moisture fluxes in the soil, including advective

heat fluxes and stable isotopes of water (not used here). In contrast to the standard CABLE soil model, SLI solves for the coupled energy moisture fluxes at the air/soil interface:

$$R_{net,soil} = \frac{\rho_a c_p}{r_{bh}}\left(T_{surface} - T_c\right) + \lambda \min\left[E_{pot}, E_{vap} + E_{liq}\right] + \frac{k_{H,1}}{\Delta z_1/2}\left(T_{surface} - T_{soil,1}\right) \quad (25)$$

The net radiation absorbed by the soil $R_{net,soil}$ [W m$^{-2}$] is calculated as in the standard CABLE2.0, except that we use the temperature at the soil/air interface (and not the temperature of the top soil layer $T_{soil,1}$) to represent the surface

temperature $T_{surface}$. On the right hand side of Equation (25), the first term is the sensible heat flux ($H_{soil}$), with $r_{bh}$ the resistance to sensible heat transfer [s m$^{-1}$]. The third term is the conduction of heat into the soil, with $k_{H,1}$ the thermal conductivity of the top soil layer [W m$^{-1}$ K$^{-1}$]. The second term is the latent heat of soil evaporation, with $E_{pot}$ the soil evaporation at a surface relative humidity of one; and $E_{vap}$ and $E_{liq}$ are the vapour and liquid components of the moisture fluxes [kg m$^{-2}$ s$^{-1}$] from within the soil column to the surface:

$$E_{pot} = \frac{\rho_a c_p \left(D_a\left[k_{H,1} r_{bh} + 0.5\Delta z_1 \rho_a c_p\right] + r_{bh} s(T_c)\left[0.5\Delta z_1 R_{net,soil} + k_{th}\left(T_{soil,1} - T_c\right)\right]\right)}{\left(c_p/\lambda\right) r_{bw}\left(k_{th} r_{bh} + 0.5\Delta z_1 \rho_a c_p\right) + 0.5\Delta z_1 r_{bh} \rho_a c_p s(T_c)} \quad (26)$$

$$E_{vap} = \frac{h_{r,1} c_{v,sat}(T_1) - c_{v,a}}{r_{b,w} + \left(\Delta x_1 / 2\right)/ D_{v,1}} \quad (27)$$

$$E_{liq} = \rho_w \left[\frac{\phi_l(h_{r,1}) - \phi_{min}}{\Delta x_1 / 2} - K_1\right] \quad (28)$$

where $D_a$ is the humidity deficit [m$^3$(H$_2$O) m$^{-3}$(air)] in the canopy; $r_{b,w}$ is the resistance to water vapour transfer [s m$^{-1}$]; $s$ is the slope of saturated vapour pressure with respect to temperature [m$^3$(H$_2$O) m$^{-3}$(air) K$^{-1}$]; $h_{r,1}$ is the relative humidity in the

top soil layer, $c_{v,sat}$ is the saturated vapour concentration [m$^3$(H$_2$O) m$^{-3}$(air)], $D_{v,1}$ is the vapour diffusivity in the top soil layer [m$^2$ s$^{-1}$]; $\phi_l$ is the liquid matric flux potential [m$^2$ s$^{-1}$]; $K_1$ is the hydraulic conductivity of the top soil layer [m s$^{-1}$]; $\phi_{min}$ [m$^2$ s$^{-1}$] is the matric flux potential corresponding to minimum soil moisture potential, set here to $h_{min} = -10^6$ m. $E_{pot}$ comes from the solution of the coupled energy and moisture conservation equations at the soil-air interface with relative humidity at the surface set to 1 (Haverd and Cuntz, 2010; Haverd et al., 2013).



**Improved parameterization of in-canopy resistance to turbulent transfer**

We adapt the CABLE2.0 formulation of $r_{soil}$ to account for local (in-canopy) stability effects on the resistance of transfer from the ground to the canopy air-space, effectively increasing the resistance when ground sensible heat fluxes are negative. The adaptation splits the resistance into the sum of two components: the first $r_{soil,a}$ from the soil roughness height to a shear height $z_{sh}$, and the second $r_{soil,b}$ from $z_{sh}$ to the displacement height $d$. We assume that the shear height, representing the depth of the shear-driven surface layer that forms along the ground surface under the canopy, is a small fraction of the canopy height, here 0.1. Both resistance components, like the original $r_{soil}$, (Equation (17)) are integrals over the inverse of the Eddy diffusivity $K_f$:

$$r_{soil,a} = \int_{z_{0s}}^{z_{sh}} \frac{dz}{K_f(z)} \tag{29}$$

$$r_{soil,b} = \int_{z_{sh}}^{d} \frac{dz}{K_f(z)} \tag{30}$$

where alternate forms of the Eddy diffusivity are specified, the first accounting for local stability effects, and the second is the same as in the original formulation of $r_{soil}$:

$$K_f(z) = \begin{cases} \dfrac{\kappa z \widetilde{u_*}}{\Phi_h\left(\dfrac{z}{\widetilde{L}}\right)} & , z_{0s} < z < z_{sh} \\[2em] \dfrac{1}{\sigma_w^2 \tau_L} & , z_{sh} < z < d \end{cases} \tag{31}$$

This yields

$$\begin{aligned} r_{soil,a} &= \widetilde{u_*} \int_{z_{0,s}}^{z_{sh}} \frac{\Phi_h\left(\dfrac{z}{\widetilde{L}}\right)}{\kappa z} dz \\ &= \widetilde{u_*}\left[ \ln\left(\frac{z_{sh}}{z_{0s}}\right) - \psi_h\left(\frac{z_{sh}}{\widetilde{L}}\right) + \psi_h\left(\frac{z_{0s}}{\widetilde{L}}\right) \right] \end{aligned} \tag{32}$$

and

$$r_{soil,b} = \frac{1}{u_*} \ln\left(\frac{d}{z_{sh}}\right) \frac{\exp\left(2c_{s,w} L\right)(d/h)}{a_3^2 c_{TL}} \tag{33}$$





In Equations (31)-(33), $\kappa$ is the von Karman constant (0.4), $\Phi_h$ is the Monin-Obukhov stability function (Garratt, 1992), $\widetilde{u_*}$ is the friction velocity at height $z_{sh}$ and is related to the friction velocity at the reference height above the canopy by the same factor that attenuates the mean windspeed in the canopy:

$$\widetilde{u_*} = u_* \exp\left\{ -c_u \left( 1 - \frac{z_{sh}}{h} \right) \right\} \qquad (34)$$

where $c_u$ is the exponent for an assumed exponential wind profile (Raupach, 1994). $\widetilde{L}$ is the local Obukhov length, correspondingly given by:

$$\widetilde{L} = \frac{-\widetilde{u}^3}{\kappa \dfrac{g}{T_K} \dfrac{H_{soil}}{\rho_a c_p}} \qquad (35)$$

where $g$ is the gravitational constant and $T_K$ is the canopy air temperature [K].

### 3 Data

Following the PLUMBER land surface model benchmarking experiment described by Best et al. (2015), we use data from 18 Eddy covariance flux tower sites, available as part of the FLUXNET LaThuile free fair-use subset (fluxdata.org; see Acknowledgements). Best et al. (2015) selected sites for broad coverage of vegetation types and climate, and we use the same sites here, with the exception of five omissions (ElSaler and ElSaler2 (irrigated); Loobos (missing GPP observations), Palang (poor energy closure) and Merbleue (wetland site)), and three inclusions (Roccarespampani,

Tharandt and Castelporzanio), such that our list of sites includes all 5 sites employed by De Kauwe et al. (2015) for their assessment of CABLE drought response during the 2003 European heatwave. Gap-filling and quality control were applied, as described by Best et al. (2015). Fluxes were aggregated to monthly and daily values for comparison with model output.

FLUXNET site locations, IGBP plant functional type and data duration are listed in Table 2, combining information from

Best et al. (2015) and De Kauwe et al. (2015).

**Table 2: List of FLUXNET site locations**

| Name | Country | Lat | Lon | CABLE PFT | Duration |
|---|---|---|---|---|---|
| Amplero | Italy | 41.90 °N | 13.61 °E | C3 Grassland | 2003-2006 |
| Blodgett | United States | 38.90 °N | 120.63 °W | Evergreen Needleleaf | 2000-2006 |





| Bugac | Hungary | 46.69 °N | 19.60 °E | C3 Grassland | 2002-2006 |
|---|---|---|---|---|---|
| Castelporziano | Italy | 41.70 °N | 12.37 °W | Evergreen Broadleaf | 2001-2006 |
| Espirra | Portugal | 38.64 °N | 8.60 °W | Evergreen Broadleaf | 2001-2006 |
| Fort Peck | United States | 48.31 °N | 105.10 °W | C3 Grassland | 2000-2006 |
| Harvard | United States | 42.54 °N | 72.17 °W | Deciduous Broadleaf | 1994-2001 |
| Hesse | France | 48.67 °N | 7.06 °E | Deciduous Broadleaf | 1999-2006 |
| Howard | Australia | 12.49 °S | 131.15 °E | C4 Grassland | 2002-2005 |
| Howlandm | United States | 45.20 °N | 68.74 °W | Evergreen Needleleaf | 1996-2004 |
| Hyytiala | Finland | 61.85 °N | 24.29 °E | Evergreen Needleleaf | 2001-2004 |
| Kruger | South Africa | 25.02 °S | 31.50 °E | C4 grassland | 2003-2004 |
| Mopane | Botswana | 19.92 °S | 23.56 °E | C4 Grassland | 199-2001 |
| Roccarespampani | Italy | 42.40 °N | 11.92 °W | Deciduous Broadleaf | 2002-2006 |
| Sylvania | United States | 46.24 °N | 89.35 °W | Deciduous Broadleaf | 2002-2005 |
| Tharandt | Germany | 58.97 °N | 13.57 °W | Evergreen Needleleaf | 1998-2005 |
| Tumbarumba | Australia | 38.66 °S | 148.15 °E | Evergreen Broadleaf | 2002-2005 |
| University Michigan | United States | 48.56 °N | 84.71 °W | Deciduous Broadleaf | 1999-2003 |

## 4 Simulations

For each site, CABLE2.0 was run using local half-hourly meteorology from the flux tower. Model soil and vegetation parameters were held fixed at their default values for the site PFT and CABLE's $1°x1°$ gridded soil texture. Leaf area index was prescribed using a $1°x1°$ gridded monthly climatology from the MODIS Collection 5 product (Ganguly et al.,

2008). Model runs were initialized by repeated forcing with site data until soil moisture and temperature convergence were achieved.

For each site, four simulations, distinguished by model configuration were performed: (i) the standard CABLE2.0 model ("STD"); (ii) the standard CABLE2.0 model with the new drought response ("STD_NDR"); (iii) the standard CABLE2.0 with the new drought response and litter effect on soil evaporation ("STD_NDR_LIT"); (iv) the standard CABLE2.0 with

SLI hydrology, including the local stability correction to the soil-canopy resistance ("SLI"). Note here that SLI already includes the new drought response and effects of litter on soil evaporation.

The new drought response parameterization requires a parameter, $\gamma$, which appears in the root shut-down function (Eq. (7) ) and is related to drought tolerance. We selected a single global value of $\gamma = 0.03$, which gave the best model performance, as assessed against monthly latent heat observations, over a range of values (0.01-0.12) for the SLI

configuration.





## 5 Results and Discussion

### 5.1 Evaluation against FLUXNET data

Figure 1 compares modeled monthly mean fluxes of latent heat flux ($\lambda$E), sensible heat flux (H), GPP and water use efficiency (defined here as GPP divided by ET, and filtered for observed monthly mean GPP > 0.5 g C m$^{-2}$ d$^{-1}$ and

5    monthly mean ET > 0.00 kg(H$_2$O) d$^{-1}$) for the four model configurations. Corresponding evaluation metrics are presented in Table 3. Figure 1 (STD) reveals clouds of points associated with very low latent heat fluxes and very high water use efficiencies compared with observations. This problem is largely resolved by the new drought response formulation (STD_NDR). Correspondingly, root-mean-squared-error (RMSE) is reduced from 27 to 23 W m$^{-2}$ for $\lambda$E; from 27 to 23 W m$^{-2}$ for H and from 3.4 to 2.3 g(C) kg(H$_2$O)$^{-1}$ for WUE (Table 3). Model performance is further improved with the

10    introduction of litter effects and SLI, particularly for evapotranspiration, with RMSE being further reduced from 23 to 17 W m$^{-2}$ (Table 3). The improvement in H is smaller, consistent with significant discrepancies between modeled and observed available energy (R$_{net}$, not shown), which are not expected to be resolved by the changes introduced here. Model performance for GPP is largely invariant across the four model configurations. All other metrics of Best et al. (2015) produced a consistent picture (only R$^2$ shown in Table 2).

Table 3: Evaluation metrics correlation coefficient (R$^2$) and root mean square error (RMSE) for monthly latent heat, sensible heat, GPP, and WUE predicted using four model configurations: (i) standard CABLE2.0 (STD); (ii) new drought response (STD_NDR); (iii) new drought response with litter effects on soil evaporation (STD_NDR_LIT); (iv) full Soil-Litter-Iso (SLI)

| | | STD | STD_NDR | STD_NDR_LIT | SLI |
|---|---|---|---|---|---|
| R$^2$ | $\lambda$E | 0.41 | 0.65 | 0.72 | 0.74 |
| | H | 0.58 | 0.60 | 0.63 | 0.63 |
| | GPP | 0.76 | 0.74 | 0.74 | 0.74 |
| | WUE | 0.00 | 0.06 | 0.09 | 0.08 |
| RMSE | $\lambda$E | 27.49 | 22.69 | 18.92 | 16.69 |
| | H | 27.26 | 23.01 | 20.93 | 22.17 |
| | GPP | 1.73 | 1.79 | 1.81 | 1.77 |
| | WUE | 3.39 | 2.34 | 2.26 | 2.31 |

Site-specific examples are shown as monthly scattergrams (Figure 2) and 14-day-running-mean time series (Figure 3) of $\lambda$E. As in Figure 1, the scattergrams show results for all four model configurations, while the time series in Figure 3 are presented only for the STD and SLI configurations, and include the modeled soil contribution to the latent heat fluxes. At the first site (Howard) there is a marked wet-dry seasonality. Here the STD and SLI configurations agree on the




magnitude of the wet-season latent heat flux, including the soil component. However in the dry season, the default model under-predicts the latent heat flux, while the improved model matches the observed gradual decline through the dry season. At Tumbarumba, both the new drought response and litter effects improve simulated $\lambda$E, since this site is subject to frequent periods of soil moisture deficit, and the open canopy allows high radiation fluxes at the ground, leading to

over-estimation of $\lambda$E during periods of high water availability and in the absence of litter. Evidence of this is seen in the excessive soil evaporation peaks in the STD configuration, but not the SLI configuration (Figure 3). Similarly, both the new drought response and litter effects improve simulated $\lambda$E at Roccarespampani: the STD model configuration predicts a severe decline in $\lambda$E during the 2003 drought episode, which is not seen in either the observtions or the SLI configuration (Figure 3). At Hyytiala, litter effects improve simulations in the spring, while the improved modeling of in-

canopy stability effects in SLI correct the highly negative winter latent heat fluxes produced by the other model configurations. Finally at Blodgett, we see marked improvements due to the new drought response and litter effects: the STD model shows an unrealistic summer decline in $\lambda$E, while the SLI configuration tracks the observations well. Similar to Hyytiala, the STD configuration reveals an over-prediction of $\lambda$E at the start of the growing season. This is associated with excessive soil evaporation, not seen in the SLI simulations, largely because of leaf litter effects, with further

dampening of soil evaporation in SLI by the modified resistance parameterisations (Equations (32) and (33)).

The significant effect of leaf litter on soil evaporation is anticipated. Ogée and Brunet (2002) and Gonzalez-Sosa et al. (1999; 2001) have demonstrated the importance of including litter on modelled soil evaporation in forest and agricultural ecosystems respectively, while Haverd and Cuntz (2010) demonstrated that accounting for litter improved the timing and partitioning of latent heat fluxes at the Tumbarumba flux site.

Sakaguchi and Zeng (2009) made a similar study to ours for the Community Land Model rev. 3.5 (CLM3.5; Oleson et al., 2008), testing different soil resistances, a litter layer and under-canopy stability effects. Each modification contributed differently over different regions and seasons in their simulations, which is very similar to our results for the globally distributed FLUXNET sites. The additional resistance due to a litter layer was much pronounced over semi-arid regions in CLM3.5, which is in line with our results for Tumbarumba and Roccarespampani but also with the results of Ogée and

Brunet (2002) who developed their litter layer model for a pine forest in Southern France. The stability modification was marginal in CLM3.5 but had significant effects during the dry season within dense forests. Our in-canopy stability improvement on the other hand has most effects over cold surfaces such as in Hyytiala, Finland during winter.

## 5.2 Sensitivity to drought tolerance parameter in the new drought response function

We explored a range of values (0.01–0.12) for the parameter $\gamma$, which determines the steepness of the root shut-down function of Lai and Katul (2000)(Eq. (7)).



Across the 18 Flux sites, a value of $\gamma = 0.03$ gave the best results for the SLI model configuration. The optimum value varied from site to site but with no apparent relationship to aridity or plant functional type. Further, the same was true when the data-set was reduced to the drought-affected European sites (Tharandt, Hesse, Castelporziano, Roccarespampani, Espirra) during 2003; a result which contrasts to the finding of De Kauwe et al. (2015) that drought

response decreases along the mesic-xeric gradient spanned by these sites. For the present study we therefore maintain a spatially invariant value of $\gamma$.

The drought response function proposed here, which depends on the soil moisture content of the wettest accessible soil layer, is designed to emulate optimal water resource use within the confines of the existing CABLE2.0 state variables. In CABLE, the only state variable available to inform root water uptake is the volumetric soil moisture content of each of the

6 soil layers. In this context, the parameterization of coupled drought response and root water extraction proposed here represents a parsimonious alternative to more mechanistic approaches in which the mechanisms being modeled require more information than is available. For example both the parameterisation of hydraulic redistribution of Ryel et al. (2002) and the root-water extraction profile of Gardner (1960) as implemented in CABLE by Li et al. (2012) and De Kauwe et al. (2015) respectively require root surface conductance, which is not represented in CABLE. Further, access to deep water

via these mechanisms is likely over-represented to compensate for assumption of a static PFT-dependent root density distribution: in reality rooting depths may be much lower than suggested by the average profiles assumed in CABLE (e.g. Canadell et al., 1996), and root density profiles are dynamic, adapting to resource availability (e.g. Haverd et al., 2016; Schymanski et al., 2009).

### 5.3 Alternative Drought Response Mechanism

There is current discussion about the mechanism by which soil moisture deficit impacts plant productivity: via stomatal conductance or via the photosynthetic apparatus, or both (e.g. Piayda et al., 2014; Zhou et al., 2013). In light of this we conducted an experiment using the SLI model configuration, modified such that the maximum rate of Rubisco activity ($V_{cmax}$) and the potential rate of electron transport ($J_{max}$) were reduced by the drought response factor $f_{w,soil}$, while the drought response of stomatal conductance was disabled. Optimum results were obtained with the same value of $\gamma = 0.03$,

and corresponding model performance varied remarkably little compared with the drought response being applied to stomatal conductance (results not shown). This experiment was not conducted to inform the mechanistic debate, but rather to illustrate that our model improvements are robust to changes in parameterisations such as this.

### 6. Conclusion

We have presented formulations for improved plant drought response and soil surface energy balance in CABLE 2.0. The equations presented here for root water extraction and stomatal drought response are not uniquely valid formulations, although they are parsimonious (requiring a single parameter) and aid in producing skillful simulations of ET at globally




distributed FLUXNET sites. What is particularly important about the model improvements presented here is that stomatal drought response and root water extraction are properly coupled such that over-extraction cannot occur and coupling between photosynthesis and transpiration is maintained, thus avoiding implausible water use efficiencies produced by the standard CABLE2.0 model configuration. Such model improvements can only be meaningfully tested against

observational estimates of total ET if soil evaporation is accurately modeled. We have shown that a physically accurate description of soil evaporation available via the SLI soil model significantly enhances predictions of total ET compared to the standard soil model in CABLE, in which supply-limited evaporation is an empirical function of upper layer soil moisture (Equations (10)-(12)), and tends to be over-estimated, particularly in the absence of litter effects.

## Acknowlegdements

This work used Eddy covariance data acquired by the FLUXNET community and in particular by the following networks: AmeriFlux [U.S. Department of Energy, Biological and Environmental Research,Terrestrial Carbon Program (DE-FG02-04ER63917 and DE-FG-02-04ER63911)], AfriFlux, CarboAfrica, CarboEuropeIP, CarboItaly, CarboMont, ChinaFlux, FLUXNET-Canada (supported by CFCAS, NSERC, BIOCAP, Environment Canada, and NRCan), GreenGrass, KoFlux,

LBA, NECC, OzFlux, TCOSSiberia, and USCCC. We acknowledge the financial support to the Eddy covariance data harmonization provided by CarboEuropeIP, FAO-GTOS-TCO, iLEAPS, Max Planck Institute for Biogeochemistry, the National Science Foundation, Tuscia University, Université Laval and Environment Canada, and the U.S. Department of Energy, and the database development and technical support from Berkeley Water Center; Lawrence Berkeley National Laboratory; Microsoft Research eScience; Oak Ridge National Laboratory; University of California, Berkeley; and

University of Virginia.

Vanessa Haverd's contribution was made possible by funding from the Australian Climate Change Science Program.

## Code Availability

The source code can be accessed after registration at https://trac.nci.org.au/trac/cable. Simulations in this work used Revision

Number 3432.

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





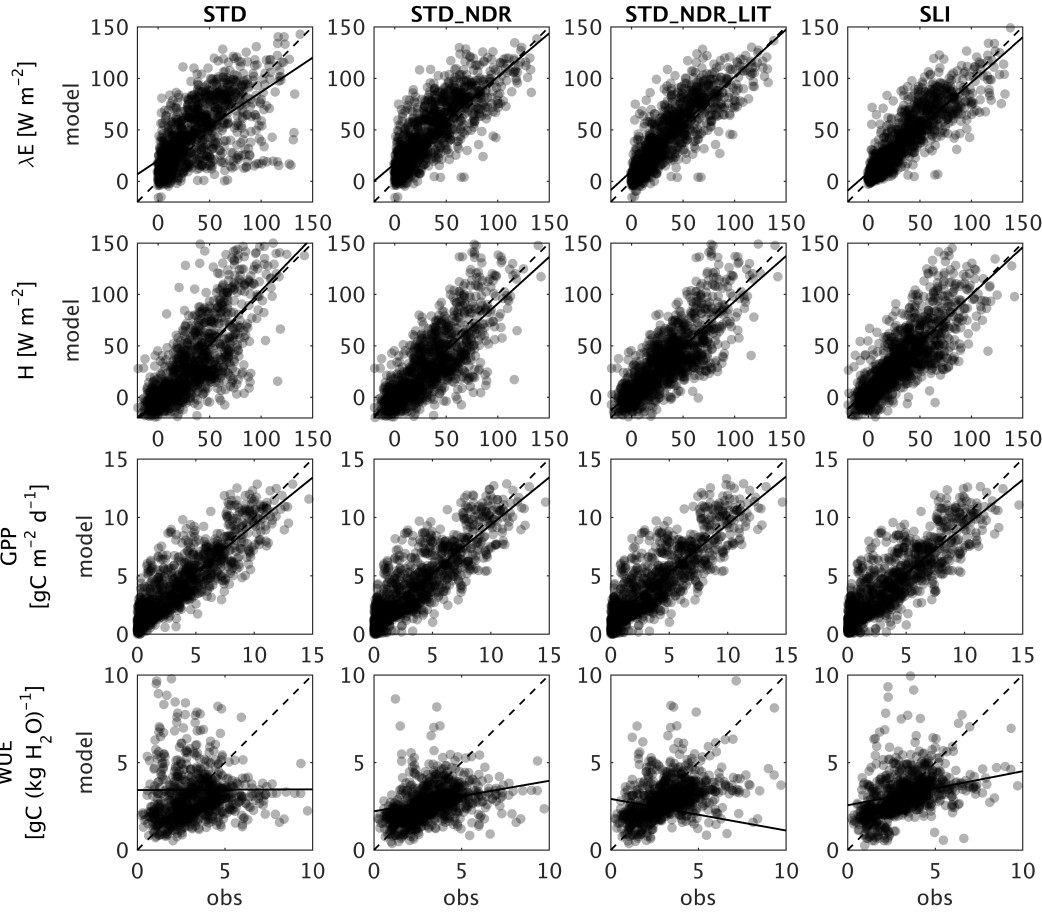

**Figure 1: Monthly modeled vs. observed latent heat, sensible heat, GPP, and total water use efficiency for four model configurations: (i) standard CABLE2.0 (STD); (ii) new drought response (STD_NDR); (iii) new drought response with litter effects on soil evaporation (STD_NDR_LIT); (iv) full Soil-Litter-Iso (SLI). Solid lines: linear regression fits; dashed lines: 1 to 1.**





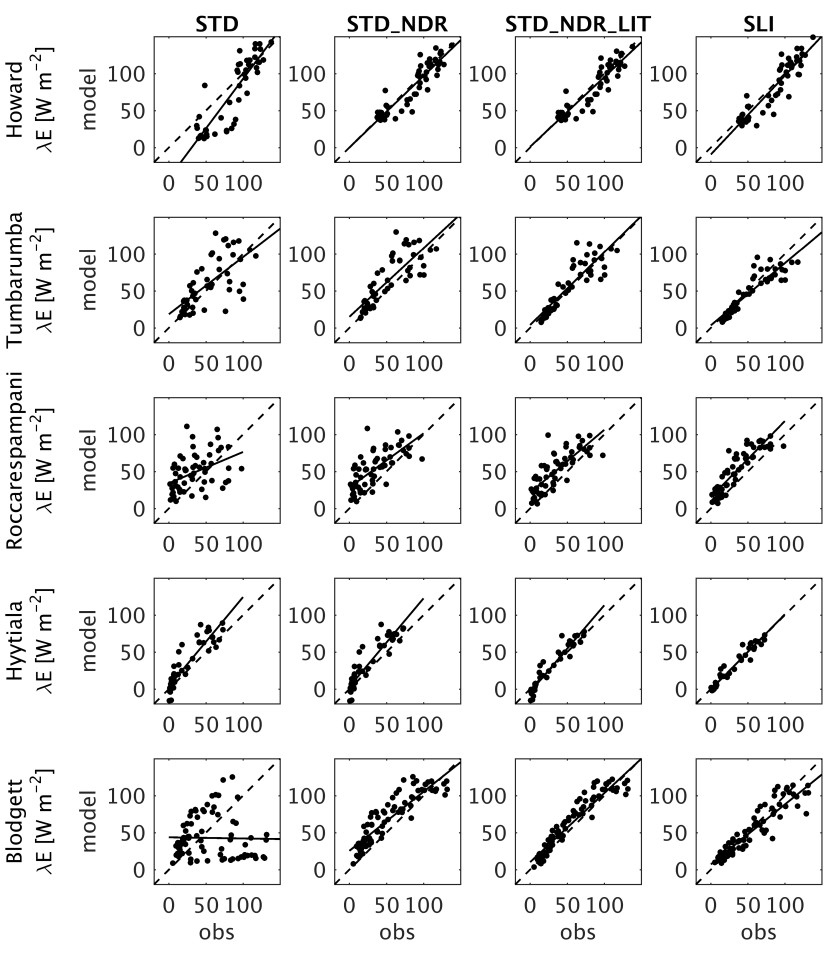

**Figure 2: Site-specific examples of monthly modeled vs. observed latent heat at 5 selected sites for four model configurations: (i) standard CABLE2.0 (STD); (ii) new drought response (STD_NDR); (iii) new drought response with litter effects on soil evaporation (STD_NDR_LIT); (iv) full Soil-Litter-Iso (SLI). Solid lines: linear regression fits; dashed lines: 1 to 1.**





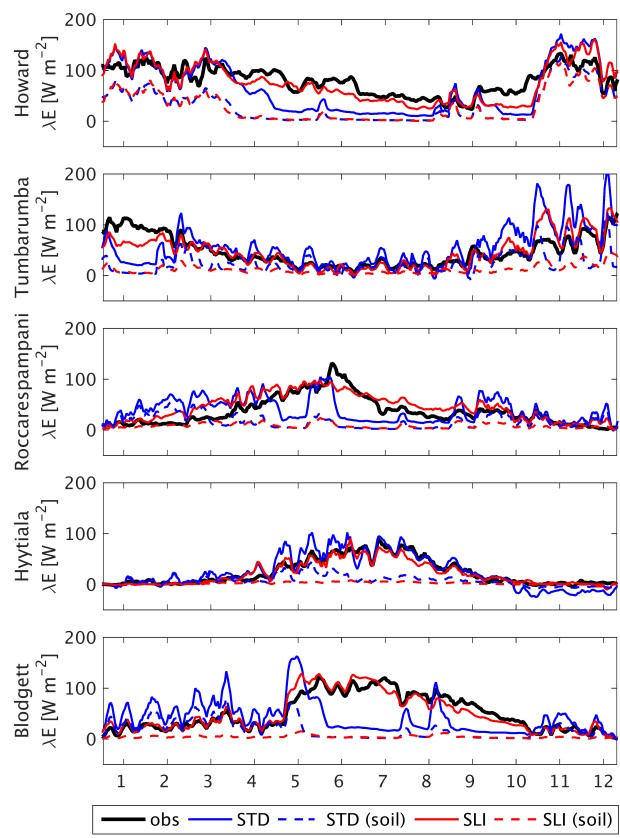

**Figure 3: Illustrative 1-y (2003) time-series of 14-day running mean modeled and observed latent heat at 5 selected sites, for two model configurations: (i) standard CABLE2.0 (STD); (ii) full Soil-Litter-Iso (SLI). Modelled soil components are shown as well.**

