# Peer review of "Improved representations of coupled soil-canopy processes in the CABLE land surface model (SVN r3432)"

_Geoscientific Model Development, 2016_

## Referee Comment (RC1) · H. Zheng (Referee) · 28 Mar 2016

This paper is representing several improvements to the coupled soil-canopy processes in the CABLE land surface model, especially a single-parameter drought response function to solve the decoupling of transpiration and photosynthesis fluxes under drying soil conditions. These improvements are important, and the estimations of the terrestrial carbon budgets and simulations of the ecosystem's response to drought events would greatly benefit from this work.

The paper is well-written and clearly aligned with the goals of the Geoscientific Model Development Journal. I recommend its publication subject to some questions on the technical details.

[Figure]

1. Eq. (7) differs from original root shut-down function of Lai and Katul (2000). In the original function, $\alpha_1(\theta)$ is a product of two items and its value will be 1 if the soil is saturated. However, Eq. (7) does not adhere to this feature, that does not seem reasonable. Why?

2. Would Eq. (8) experience a "division by zero" error? How to avoid this error?

3. Eq. (4): Is the coefficient 1.1 necessary, while $(\theta_j - \theta_w)\Delta z_j$ represents the water available in the $j^{th}$ soil layer?

4. Eq. (14): Please describe the variable $c_{sw}$? Also in Eq. (17) and Eq. (33).

5. Eq. (28): Please describe the variable $\Delta x_1$?

Some specific comments:

1. Page 1, Line 19: global Eddy covariance flux network → global eddy-covariance FLUX NETwork.

2. Page 3, line 10: (2011)(CABLE1.4b) → (2011) (CABLE1.4b). Missing space.

3. Page 9, line 13: *(21)* → (21). Italic fonts.

4. Page 9, line 17: $2.38\ m\ s$-1 or $2.38\ m$-1 $s$? Please check the unit. Also please correct the minus signs on this line.

5. Page 10, line 22: $h_{min} = 10^{-6}$ or $\phi_{min} = 10^{-6}$? Please check it.

6. Page 13, line 14: 0.01-0.12 → 0.01–0.12. hyphen → en dash.

---

## Referee Comment (RC2) · E. Blyth (Referee) · 19 Apr 2016

This paper presents a thorough examination of the impact of various improvements to the CABLE land surface model. Covering several really interesting aspects of land surface modelling: the drought response of the vegetation through their roots, the aerodynamics of the canopy and its effect on the energy and water balance and the addition of a leaf litter layer to inhibit the evaporation from the bare soil. All of these aspects need improving - probably in many of the current land surface models - and it is really interesting to see a paper lay all these out and then check the performance against some data. I guess the only thing missing is to see the performance checked when it is run in coupled made - but I suppose that is the task of a different paper. This one is really setting the scene and explaining the changes to the model. I think it suits the journal well and it will be of interest to many readers - both users of the CABLE model

and to other modellers. It is also good to see the data used in an intelligent way.

---

## Referee Comment (RC3) · Anonymous Referee #3 · 25 Apr 2016

**Review of "Improved representations of coupled soil-canopy processes in the CABLE land surface mode" by Haverd et al.**

The paper documents a solution to a known issue with the CABLE LSM, namely that CABLE simulates un-realistically high WUE (GPP/ET), even under drought conditions. This is fixed by 3 different changes to the code. The paper fits very well within the scope of GMD. The paper should be considered for publication in GMD after the following comments have been addressed:

**Major comments:**

1. The introduction needs a bit more clarity, especially for the non-CABLE expert. At page 2, line 25, it is stated that Haverd et al. (2013) implemented an alternative formulation for coupled drought response and root water extraction in CABLE (no version is mentioned). At line 32, it is mentioned that one of the aims of the paper is to implement the new scheme from Haverd et al. (2013) in CABLE2.0. This reads like you are repeating work already done, as you do not explain that the version of CABLE used by Haverd et al. (2013) is for BIOS2, and this is Not the version of CABLE current used in ACCESS as we speak. A non-CABLE expert will be left confused if you don't explain this a bit more.
2. The work of Li et al. (2012) and De Kauwe et al. (2015) needs to be better explained and put into context of this paper. Namely, how does this current paper differ from the previous two, since these also addressed broadly the same issue in CABLE. My understanding of the work of Li et al. (2012) is that it was at a single site, and this work cannot be generalized when running CABLE as a global model, whereas your can be. I think you should make this clearer. Also, how are you building/improving on De Kauwe et al. (2015), it is not clear. The latter addressed broadly the same issue. So, how is your paper different?
3. Provide the reference for this "data on maximum vegetation rooting depth" at page 2, line 33.
4. Page 3, lines 1 to 5, the second and the third aim both relate to the implementation of the SLI model in CABLE. How are these two distinct aims? This paper is documenting not one, but two major code changes to CABLE, the new drought response, as well as the SLI model. Hence, a lot more information/context should be provided about SLI in the introduction. You leave the reader with many questions and clarification is needed on all 3 stages on development, why each one is needed on its own, and why the combination of all 3 is necessary to fix this issue in CABLE.
5. In the description of Canopy photosynthesis, it should be noted that CABLEv2.0 has a new Stomatal Conductance Scheme, which is an improvement on the default scheme, as documented by De Kauwe et al. (2015), Kala et al. (2015), Kala et al. (2016). Almost all future simulations within ACCESS are likely to use the new scheme, rather than the default, hence this is worth noting.

6. Equations 16 and 17, it is simply stated that a different integration of Eq. 13 is used, as compared to the default, without any explanation(s) and leaves the reader wondering.
7. Page 9, lines 5 to 10 – Clitt parameter values are obtained by separate offline spin-up using GSWP2-3 forcing. Firstly, which one did you use? GSWP2 or GSWP3? Or both? If both, then did you take the average from the two? Did you run CABLE offline globally with GSWP2/3, then take the average over all PFTs? Or did you extract single site forcing from GSWP and run single-site offline simulations? Secondly, these parameters are therefore model dependant, namely CASA-CNP, rather than have any direct link to observations. This is not discussed at all. This is parameter tuning, and you need to make this explicit and flag the implications.
8. This paper has 35 equations in total within the main text, and the reader feels rather dazzled after going through all 35! I strongly recommend moving some of these to an Appendix, and focus only on the relevant equations.
9. A map showing the locations of the 18 sites would be good.
10. Page 13, lines 12-15, the tuning of the parameter, gamma, is suddenly introduced. This parameter is used in Eq. 7, which is from Lai and Katul (2000). There is no discussion if this parameter value of 0.03 obtained from tuning is different to the value used by Lai and Katul (2000) or any other study? This is just presented without any context and leaves the reader wondering.
11. Additionally, there needs to be a discussion about the parameter tuning carried out in this study (offline only) and what the implications would be for coupled (ACCESS) simulations. Would one simply used the same parameter values for coupled simulations?
12. Figure 1 – This is no explanation of how the black circles differ from the grey ones? It seems to me that the main improvement is in latent heat, very little in GPP, so the main improvement in WUE is due to latent. The simulation of latent heat is largely improved. This is a great achievement.
13. Table 3 – The bias (model – obs) should be added.
14. Page 16, line 4, by "contrasts", you mean contradictory? If yes, then some more in-depth discussion of why would seem appropriate.
15. I was rather surprised that the authors did not conduct or show any results from Global offline simulations using GSWP2 or GSWP3, especially, given that they used GSWP2/3 to tune some parameters. To better inform the use of these modifications in CABLE when coupled to ACCESS, global offline simulations are extremely valuable, and would make a very useful addition to this paper (rather short with only 3 figures). Additionally, other studies which have tested new developments to CABLE have used both single site and global offline GSWP simulations (De Kauwe et al. (2015) and evaluated CABLE's ET against gridded observational products such as LandFlux data. This study should present some global offline results using GSWP2 or GSWP3.
16. This study make No mention of the fact that CABLEv2.0 now has a new, improved and more physically realistic hydrology parameterization, as described in detail by Decker et al. (2015). The new hydrology makes significant improvements to CABLE excessive ET. Whilst it is well outside

the scope of this paper to test the current modifications with the new hydrology by Decker et al. (2015), this must be explicitly discussed as critical future work which needs to be carried out.

Decker, M. (2015), Development and evaluation of a new soil moisture and runoff parameterization for the CABLE LSM including subgrid-scale processes, J. Adv. Model. Earth Syst., 7, 1788–1809, doi:10.1002/2015MS000507.

De Kauwe, M. G**.,** Kala, J., Lin, Y.-S., Pitman, A. J., Medlyn, B. E., Duursma, R. A., Abramowitz, G., Wang Y.-P., and Miralles D. G. (2015) A test of an optimal stomatal conductance scheme within the CABLE Land Surface Model. *Geoscientific Model Development*, 8, 431-452, doi: 10.5194/gmd-8-431-2015

Kala, J., M. G. De Kauwe, A. J. Pitman, R. Lorenz, B. E. Medlyn, Y.-P Wang, Y.-S Lin, and G. Abramowitz (2015) Implementation of an optimal stomatal conductance scheme in the Australian Community Climate Earth Systems Simulator (ACCESS1.3b). *Geoscientific Model Development*., 8, 3877-3889, doi:10.5194/gmd-8-3877-2015

Kala, J., M. G. De Kauwe, A. J. Pitman, R. Lorenz, B. E. Medlyn, Y. P. Wang, and S. E. Perkins-Kirkpatrick (2016) Impact of the representation of stomatal conductance on model projections of heatwave intensity. *Scientific Reports,* 6, 23418, doi:10.1038/srep23418

---

## Author Comment (AC1) · 10 Jun 2016

Pleas see "figure1-pdf" for revised manuscript.

Please also note the supplement to this comment:
http://www.geosci-model-dev-discuss.net/gmd-2016-37/gmd-2016-37-AC1-supplement.pdf

**Improved representations of coupled soil-canopy processes in the CABLE land surface model**

Vanessa Haverd[1], Matthias Cuntz[2], Lars P. Nieradzik[1], Ian N. Harman[1]

[1] CSIRO Oceans and Atmosphere, P.O. Box 3023, Canberra ACT 2601, Australia.

[2] Department Computational Hydrosystems, UFZ—Helmholtz Centre for Environmental Research, Permoserstr. 15, 04318 Leipzig, Germany

*Correspondence to*: Vanessa Haverd (Vanessa.haverd@csiro.au)

**Abstract.** CABLE is a global land surface model, which has been used extensively in offline and coupled simulations. While CABLE performs well in comparison with other land surface models, results are impacted by decoupling of transpiration and photosynthesis fluxes under drying soil conditions, often leading to implausibly high water use efficiencies. Here we present a solution to this problem, ensuring that modeled transpiration is always consistent with modeled photosynthesis, while introducing a parsimonious single-parameter drought response function which is coupled to root water uptake. We further improve CABLE's simulation of coupled soil-canopy processes by introducing an alternative hydrology model with a physically accurate representation of coupled energy and water fluxes at the soil/air interface, including a more realistic formulation of transfer under atmospherically stable conditions within the canopy and in the presence of leaf litter. The effects of these model developments are assessed using data from 18 stations from the global eddy-covariance FLUX NETwork, selected to span a large climatic range. Marked improvements are demonstrated, with root-mean-squared errors for monthly latent heat fluxes and water use efficiencies being reduced by 40%. Results highlight the important roles of deep soil moisture in mediating drought response and litter in dampening soil evaporation.

**Fig. 1.**

**Supplement:**

**Reviewer 1 (H Zheng)**

This paper is representing several improvements to the coupled soil-canopy processes in the CABLE land surface model, especially a single-parameter drought response function to solve the decoupling of transpiration and photosynthesis fluxes under drying
soil conditions. These improvements are important, and the estimations of the terrestrial
carbon budgets and simulations of the ecosystem's response to drought events would greatly benefit from this work.
The paper is well-written and clearly aligned with the goals of the Geoscientific Model
Development Journal. I recommend its publication subject to some questions on the technical details.

**Comment 1.1.**

Eq. (7) differs from original root shut-down function of Lai and Katul (2000). In the original function, $\alpha(\theta)$ is a product of two items and its value will be 1 if the soil is saturated. However, Eq. (7) does not adhere to this feature, that does not seem reasonable. Why?

**Response 1.1.**

We have clarified the difference between our formulation and Lai and Katulas follows (p6 L14-19):

"Note that while the functional form of Equation (7) is taken from Lai and Katul (2000), there is not a direct equivalence of parameter values because of its different implementation here.. In particular, we use the root "shut-down" function to determine stomatal drought response via Equation (9),whereas Lai and Katul (2000) multiply it by a "maximum efficiency" function, which is in turn scaled by local root density and potential evaporation to obtain actual root water extraction."

The reviewer is right that Equation (7) doesn't equal one at saturation , but it is very close (typical values 0.95-0.97). One could rescale the function to equal one at saturation but ,after retuning gamma, this would have negligible impact on results.

**Comment 1.2.**

Would Eq. (8) experience a "division by zero" error? How to avoid this error?

**Response 1.2**

We have modified Equation (8) to account for the condition when the denominator is zero.

**Comment 1.3. Eq. (4): Is the coefficient 1.1 necessary, while $\left(\theta_j - \theta_w\right)\Delta z_j$ represents**

the water available in the jth soil layer?

**Response 1.3**
Yes, the reviewer is correct. However this equation is part of the standard model configuration, and is therefore required as it is to describe the formulation of the model prior to our impriovements.

**Comment 1.4.**
Eq. (14): Please describe the variable csw? Also in Eq. (17) and Eq. (33).

**Response 1.4**
We have included this information (P9, L3):
"$c_{sw}$ is a constant determining the rate of decrease of $\sigma_w$ with depth in the canopy, with value set to 1.0."

**Comment 1.5**.
Eq. (28): Please describe the variable $\Delta x1$?

**Response 1.5:** This should be $\Delta z_1$. We have corrected the Equation.

**Some specific comments:**

**Comment 1.6**.
 Page 1, Line 19: global Eddy covariance flux network ! global eddy-covariance FLUX NETwork.

**Response 1.6**
Done

**Comment 1.7**
Page 3, line 10: (2011)(CABLE1.4b)  (2011) (CABLE1.4b). Missing space.

**Response 1.7**
Done

**Comment 1.8**
 Page 9, line 13: (21)  (21). Italic fonts.

**Response 1.8**
 Done

**Comment 1.9.**
Page 9, line 17: 2:38 m s-1 or 2:38 m-1 s? Please check the unit. Also please correct the minus signs on this line.

**Response 1.9**

$2{:}38\ \mathrm{m}^{-1}$ s: we have corrected this.

**Comment 1.10**
Page 10, line 22: hmin = 10-6 or _min = 10-6? Please check it.

**Response 1.10**
$h_{min}$ is intended.

**Comment 1.11**
Page 13, line 14: 0.01-0.12 0.01–0.12. hyphen en dash.

**Response 1.12**
Done

**Reviewer 2 (E Blyth)**

**Comment 2.1**
This paper presents a thorough examination of the impact of various improvements to the CABLE land surface model. Covering several really interesting aspects of land surface modelling: the drought response of the vegetation through their roots, the aerodynamics of the canopy and its effect on the energy and water balance and the addition of a leaf litter layer to inhibit the evaporation from the bare soil. All of these aspects need improving - probably in many of the current land surface models - and it is really interesting to see a paper lay all these out and then check the performance against some data. I guess the only thing missing is to see the performance checked when it is run in coupled mode - but I suppose that is the task of a different paper. This one is really setting the scene and explaining the changes to the model. I think it suits the journal well and it will be of interest to many readers - both users of the CABLE model and to other modellers. It is also good to see the data used in an intelligent way.

**Response 2.1**
Thank-you for the positive comments.

**Reviewer 3**

The paper documents a solution to a known issue with the CABLE LSM, namely that CABLE simulates un-realistically high WUE (GPP/ET), even under drought conditions. This is fixed by 3 different changes to the code. The paper fits very well within the scope of GMD. The paper should be considered for publication in GMD after the following comments have been addressed:

**Comment 3.1**

The introduction needs a bit more clarity, especially for the non-CABLE expert. At page 2, line 25, it is stated that Haverd et al. (2013) implemented an alternative formulation for coupled drought response and root water extraction in CABLE (no version is mentioned). At line 32, it is mentioned that one of the aims of the paper is to implement the new scheme from Haverd et al. (2013) in CABLE2.0. This reads like you are repeating work already done, as you do not explain that the version of CABLE used by Haverd et al. (2013) is for BIOS2, and this is Not the version of CABLE current used in ACCESS as we speak. A non-CABLE expert will be left confused if you don't explain this a bit more.

**Response 3.1**

We have clarified the transfer of paramterisations from the Austtralian regional application to the global context as follows (p3 L7-10):
"In this work, we take lessons learnt from the Australian regional application (Haverd et al., 2013)  and apply them globally. In particularl, we seek to resolve in CABLE 2.0 the problems of over-sensitivity of ET to drought and decoupling of transpiration and photosynthesis fluxes under drying soil conditions."

**Comment 3.2**

The work of Li et al. (2012) and De Kauwe et al. (2015) needs to be betterexplained and put into context of this paper. Namely, how does this current paper differ from the previous two, since these also addressed broadly the same issue in CABLE. My understanding of the work of Li et al. (2012) is that it was at a single site, and this work cannot be generalized when running CABLE as a global model, whereas your can be. I think you should make this clearer. Also, how are you building/improving on De Kauwe et al. (2015), it is not clear. The latter addressed broadly the same issue. So, how is your paper different?

**Response 3.2:**

We have clarified (p2 L17-18):
"The responses of gross primary production (GPP) and evapotranspiration (ET) to soil water availability in CABLE have featured in recent studies by Li et al. (2012) and De Kauwe et al. (2015a), who both considered a limited number of locations (3 and 5 respectively)."

**Comment 3.3.**

Provide the reference for this "data on maximum vegetation rooting depth" at page 2, line 33.

**Response 3.3.**

Done

**Comment 3.4.**

Page 3, lines 1 to 5, the second and the third aim both relate to the implementation of the SLI model in CABLE. How are these two distinct aims? This paper is documenting not one, but two major code changes to CABLE, the new drought response, as well as the SLI model. Hence, a lot more information/context should be provided about SLI in the introduction. You leave the reader with many questions and clarification is needed on all 3 stages on development, why each one is needed on its own, and why the combination of all 3 is necessary to fix this issue in CABLE.

**Response 3.4.**

We have clarified the rationale for including SLI in the series of model configurations as follows (P3 L14-18):
"By default, SLI includes the alternative drought response and litter effects. In contrast to the standard model configuration, it also represents coupled heat and moisture fluxes within the soil column and at the soil-air interface, and newly accounts for local stability effects on the resistance of transfer from the ground to the canopy air-space."

**Comment 3.5**

In the description of Canopy photosynthesis, it should be noted that CABLEv2.0 has a new Stomatal Conductance Scheme, which is an improvement on the default scheme, as documented by De Kauwe et al. (2015), Kala et al. (2015), Kala et al. (2016). Almost all future simulations within ACCESS are likely to use the new scheme, rather than the default, hence this is worth noting.

**Response 3.5**

This work has now been referenced in the introduction (P2 L26-28):
"Modification to the vapour-pressure deficit response of stomatal conductance in CABLE (De Kauwe et al., 2015b, Kala et et al. 2015, Kala et al. 2016) has also featured in recent studies, but it is evident that deficiencies in the predictions of seasonal cycles of evaporation are not resolved by this modification (De Kauwe et al., 2015b; Fig 3) "

**Comment 3.6**

Equations 16 and 17, it is simply stated that a different integration of Eq. 13 is used, as compared to the default, without any explanation(s) and leaves the reader wondering.

**Response 3.6**

We have expanded the text and equations as follows (p8 ; L15 forward)
"The default model uses an approximation to the integral in Equation 13, which assumes a fixed value of $\sigma_w$ with height over the range of interest:

$$r_{soil} \simeq \frac{1}{\overline{\sigma_w^2}} \int\limits_{z0s}^{d} \frac{1}{\tau_L}$$

$$= \ln\left\{\frac{d}{z_{0S}}\right\} \frac{\exp\{2c_{sw}L\} - \exp\left\{2c_{sw}L\left(1-\frac{d}{h}\right)\right\}}{a_3^2 c_{TL} 2c_{sw}L} \tag{1}$$

where

$$\overline{\sigma_w^2} = \frac{1}{d} \int\limits_0^d \sigma_w^2 dz \tag{2}$$

as used by Raupach et al. (1997) and subsequently propagated to CABLE (Wang et al., 2011, Eq A.14). However the analytic form of the integral is (Haverd et al., 2013):

$$r_{soil} = \frac{1}{u_*} \ln\left\{\frac{d}{z_{0s}}\right\} \frac{\exp\{2c_{s,w}L\}\left(\frac{d}{h}\right)}{a_3^2 c_{TL}} \tag{3}$$

and results in higher values of $r_{soil}$."

**Comment 3.7**
Page 9, lines 5 to 10 – Clitt parameter values are obtained by separate offline spin-up using GSWP2-3 forcing. Firstly, which one did you use? GSWP2 or GSWP3? Or both? If both, then did you take the average from the two? Did you run CABLE offline globally with GSWP2/3, then take the average over all PFTs? Or did you extract single site forcing from GSWP and run single-site offline simulations? Secondly, these parameters are therefore model dependant, namely CASA-CNP, rather than have any direct link to observations. This is not discussed at all. This is parameter tuning, and you need to make this explicit and flag the implications.

**Response 3.7**
 We have clarified this (P10, L6-12):
"These were obtained by running the model for 18 FLUXNET sites (Table 2) with biogeochemistry enabled (carbon-cycle only: nitrogen- and phosphorous-cycles were disabled) using repeated GSWP-2 three-hourly meteorology for the 1986-1995 period (Dirmeyer et al., 2006) until carbon pool convergence was achieved. Values of $C_{litt}$ used here are internally consistent with the carbon-cycle enabled version of CABLE. They don't reflect observation directly and were extrapolated to PFT-specific parameter values for the purpose of simulations (such as those presented here) which don't include the carbon-cycle. However, for simulations with carbon-cycle enabled, we recommend the use of internal litter carbon pools instead."

**Comment 3.8**

This paper has 35 equations in total within the main text, and the reader feels rather dazzled after going through all 35! I strongly recommend moving some of these to an Appendix, and focus only on the relevant equations.

**Resposne 3.8**

We consider all the equations presented here to be relevant and feel it appropriate to retain them in the body of the text, particularly since this is a model development paper.

**Comment 3.9**

 A map showing the locations of the 18 sites would be good.

**Response 3.9**

We feel that the location co-ordinates in Table 2 are sufficient.

**Comment 3.10**

 Page 13, lines 12-15, the tuning of the parameter, gamma, is suddenly introduced. This parameter is used in Eq. 7, which is from Lai and Katul (2000). There is no discussion if this parameter value of 0.03 obtained from tuning is different to the value used by Lai and Katul (2000) or any other study? This is just presented without any context and leaves the reader wondering.

**Response 3.10**

Where we present the drought-response function, we now include the following qualification (p6 L14-19):

"Note that while the functional form of Equation 7 is taken from Lai and Katul (2000), there is not a direct equivalence of parameter values because of its different implementation here. In particular, we use the root "shut-down" function to determine stomatal drought response via Equation (9),whereas Lai and Katul (2000) multiply it by a "maximum efficiency" function, which is in turn scaled by local root density and potential evaporation to obtain actual root water extraction."

We now highlight that this is the only tunable parameter in the new drought response formulation, and compare with the value derived for Australian vegetation (p17 L3-8):
"We explored a range of values (0.01–0.12) for the parameter $\gamma$, which determines the steepness of the root shut-down function of Lai and Katul (2000)(Equation 7), and is the single tunable parameter in the new drought response function (Equations 7-9).

Across the 18 FLUXNET sites, a value of $\gamma = 0.03$ gave the best results for the SLI model configuration., slightly higher than the low value of $\gamma = 0.01$ (reflecting high drought-tolerance) for Australian vegetation (Haverd et al., 2013)"

**Comment 3.11**
Additionally, there needs to be a discussion about the parameter tuning carried out in this study (offline only) and what the implications would be for coupled (ACCESS) simulations. Would one simply used the same parameter values for coupled simulations?

**Response 3.11**
As state above (Response 3.10), only one new tunable parameter was introduced, and optimized. The focus of this paper is offline simulations and it is out of scope to explore the transferability of parameters to coupled simulations.

**Comment 3.12.**
Figure 1 – This is no explanation of how the black circles differ from the grey ones? It seems to me that the main improvement is in latent heat, very little in GPP, so the main improvement in WUE is due to latent. The simulation of latent heat is largely improved. This is a great achievement.

**Response 3.12**
The following text has been added to the caption of Figure 1:  "Darker shading indicates higher density of points "

**Comment 3.13**
Table 3 – The bias (model – obs) should be added.

**Response 3.13**
Bias Error is now included in Table 3.

**Comment 3.14**
Page 16, line 4, by "contrasts", you mean contradictory? If yes, then some more in-depth discussion of why would seem appropriate.

**Response 3.14**
We have clarified the different findings as follows (p17 L9-13):
"Further, the same was true when the data-set was reduced to the drought-affected European sites (Tharandt, Hesse, Castelporziano, Roccarespampani, Espirra) during 2003, as selected by De Kauwe et al. (2015a). In this respect, our results do not confirm the finding of De Kauwe et al. (2015a) that parameters representing high drought sensitivity at the most mesic sites, and low drought sensitivity at the most xeric sites, are necessary to accurately model responses during drought. "

**Comment 3.15**
I was rather surprised that the authors did not conduct or show any results from Global offline simulations using GSWP2 or GSWP3, especially, given that they used GSWP2/3 to tune some parameters. To better inform

the use of these modifications in CABLE when coupled to ACCESS, global offline simulations are extremely valuable, and would make a very useful addition to this paper (rather short with only 3 figures). Additionally, other studies which have tested new developments to CABLE have used both single site and global offline GSWP simulations (De Kauwe et al. (2015) and evaluated CABLE's ET against gridded observational products such as LandFlux data. This study should present some global offline results using GSWP2 or GSWP3.

**Response 3.15**

This would be a significant extension (details below) , which we will consider in due course, but consider to be out of scope for the current paper . Global simulations on their own would produce little advance in understanding and risk confusing the reader.

Reasons for global benchmarking being a significant extension:

1. Benchmarking/comparing the global offline simulations against global products isn't justification for advance (or not) of an LSM. Two reasons – both stem from the fact that global products are exactly that "products" of another model.

   (a): using the global products to fine tune an LSM assumes that the underpinning models are in some way congruous (i.e. outputs are exactly equivalent)

   (b) biases and errors in response in the global products get transferred into the LSM. These can come from inherent weaknesses in the global products (i.e. missing processes), the parameters used within the model deriving the global products and, importantly, the structure of the model itself.

   Consequently while comparison against global products is useful such studies have to be done with a full and careful analysis of how (if) the LSM and global product can be compared.

2. Any comparison between model-model or model-observation should be expressed within the uncertainties (error bounds) of the two sets of data. While we (may) have a handle on these issues for the means we do not have the equivalent knowledge for the extremes. For example, on the basis of using a global product could we actually definitively show that the new drought response parameterisation improves CABLE's performance?

3. As the reviewer him/herself points out this work involves 3 quasi-independent advances – each of which leads to impacts on least 3 time scales (diurnal, seasonal, interannual). Consequently any extension of the work to global offline simulations would need to cover this breakdown.

**Comment 3.16**

This study makes no mention of the fact that CABLEv2.0 now has a new, improved and more physically realistic hydrology parameterization, as described in detail by Decker et al. (2015). The new hydrology makes significant improvements to CABLE excessive ET. Whilst it is well outside the scope of this paper to test the current modifications with the new hydrology by Decker et al. (2015), this must be explicitly discussed as

critical future work which needs to be carried out.

**Response 3.16**

We now reference Decker (2015) in the introduction (P2, L28-31):
"Recently Decker (2015) introduced to CABLE new conceptual parameterizations of subgrid-scale soil moisture, runoff generation, and groundwater, and showed improved performance against observation-based estimates of global ET, without modifying CABLE's vegetation response to soil moisture."

 and in the conclusion (P18 L26-27):

"Future work will entail merging the improvements demonstrated here with the new hydrological parameterisations in CABLE  (Decker 2015), and testing against global estimates of ET and runoff."

---

## Referee Report (RR1)

CABLE is the land surface scheme in the ACCESS earth system model. This is stated by the authors themselves in the Introduction. By definition, it is critical that advancements to the CABLE model be applicable at the global scale, both offline and coupled. The logical step in model development is to go from single site offline, to global offline, to fully coupled. In response to comment 3.11, the authors state that: "it is out of scope to explore the transferability of parameters to coupled simulations". I am Not asking you to carry out coupled simulations, I am saying that you need to discuss this explicitly in the paper, rather than just in the reply. Additionally, the title should also be changed to "Improved representations of …….. CABLE land surface model in offline single site simulations".

I also would like to remind the authors that GMD policy requires model version numbers in the title:
"The main paper must give the model name and version number (or other unique identifier) in the title". For more details, please see:
http://www.geoscientific-model-development.net/about/manuscript_types.html

The response to comment 3.2 has not really answered my question. The only difference between this work and that of Li et al. (2015) and De Kauwe et al. (2015) is surely not just that the latter used 3 and 5 sites respectively. I was referring to how the approach taken by these papers is different, not just that they used fewer sites.

The response to comment 3.4 is also only partial. I asked for clarification on the 3 stages of development, but all you have done is provide some more background information on SLI.

Response 3.11 – I am NOT arguing with you that it is out of scope of this paper to include coupled simulations. I am saying that you need to make it explicitly clear in the manuscript that the introduction of this tuneable parameter(s) poses a limitation in using your modified version of CABLE in coupled simulations within ACCESS.

Response 3.15:
1. I completely disagree that according to the authors that benchmarking against global observationally derived products isn't justification for the advancement or not of an LSM. Global ET products such as LandFlux EVAL (http://www.iac.ethz.ch/group/land-climate-dynamics/research/landflux-eval.html) are an invaluable dataset which one can use to determine how the LSM compares to best available Remote Sensing estimates, as well as other LSMs. For example, Decker et al. (2015) used LandFlux data, amongst others, to benchmark CABLE ET, and showed significant improvements in CABLE ET compared to best available estimates, taking into consideration the uncertainties within these best available estimates. This is just one example of how several papers have made correct use of such products in quantifying the advancement of LSMs.

a. I never asked the authors to fine-tune CABLE to match global products. This would be pointless. Benchmarking does not mean fine-tuning a model to observations.
b. The authors claim that "Biases and errors in response in the global products gets transferred to the LSM". Sorry, but this makes no sense to me.
2. Yes, of course any comparison between model-model and model-observation should be expressed within the error bounds of the two datasets. That's Exactly why products such as LandFlux provide error bounds. Not having observationally based data on extremes, does not preclude you from using observationally based products all together. Your paper examines monthly means anyway as shown in figures 1 and 2. You paper does not focus on drought periods, so what point is being made here?
3. Yes, of course it would, and there is plenty of space in this paper to cover this analysis. You paper currently only includes 3 figures.

The reasons given as to why global offline simulations would produce little advancement are not valid in my opinion, as explained above, and I simply don't see how this would confuse the reader.

---

## Author Response (AR2)

**Topical Editor Decision: Publish subject to minor revisions (Editor review)** (12 Jul 2016) by Dr. Gerd A. Folberth

Comments to the Author:

It would like to thank both, the reviewer's and the authors for all their efforts that have contributed to this highly improved revised version of the manuscript. Pending a small number of minor changes and

additions, the manuscript is ready to be published in my view.

The remaining changes I would like to request are the following:

10 1.) I agree with the reviewer of the revised version that GMD policy indeed requires model version numbers in the title of every GMD paper. I thank the reviewer for pointing this out and apologize to the authors for not spotting it myself earlier.

**Response: done. The title now reads : "Improved representations of coupled soil-canopy processes in the CABLE land surface model (SVN r3432)"**

15

5

2.) I concur with the reviewer's remark 3.15 to the original manuscript and the reply to the response to 3.15 in the author's response. This is a valid and important point. Furthermore, I think the reviewer's request to discuss the potential implications the global model configuration - not more than a paragraph or two is required - is quite reasonable and should be included. The author's could make reference to

20 plans for such an evaluation and assessment study in the future. I think it is universally agreed that such a study would go well beyond the scope of this paper and for it to be useful, indeed, needs much effort and consideration and cannot be accomplished now.

Response: we have extended the final paragraph of the conclusion to refer to future work on global benchmarking: "Future work will entail merging the improvements demonstrated here with the new

- 25 hydrological parameterisations in CABLE (Decker 2015), and a new module for woody vegetation demography and landscape heterogeneity mediated by disturbance (Haverd et al. 2014). Global simulations will be evaluated against gridded global estimates of ET, GPP, vegetation cover, biomass, and soil carbon, as well as interannual variations in atmospheric CO2 concentration. This will provide benchmarks for the use of CABLE in global offline applications (e.g. attribution of
- 30 terrestrial carbon sink), and is a necessary step towards assessing whether the modifications lead to improvements in simulated climate when CABLE is coupled to an Earth system model."

3.) I would like to encourage the authors to follow the reviewer's suggestions in relation to remarks 3.2 and 3.4 and expand on their revisions along the lines indicated. I leave this to the author's discretion,

1

35 however.

Response: we choose not to expand further here.

[revised manuscript text omitted]

(27)

$$E_{vap} = \frac{h_{r,l}c_{v,sal}(T_l) - c_{v,a}}{r_{b,w} + (\Delta z_l / 2) / D_{v,l}}$$
(28)

$$E_{liq} = \rho_{w} \left[ \frac{\phi_{l}(h_{r,l}) - \phi_{\min}}{\Delta z_{1}/2} - K_{1} \right]$$
(29)

25 where  $D_a$  is the humidity deficit  $[m^3(H_2O) m^{-3}(air)]$  in the canopy;  $r_{b,w}$  is the resistance to water vapour transfer [s m-1]; s is the slope of saturated vapour pressure with respect to temperature  $[m^3(H_2O) m^{-3}(air) K^{-1}]$ ;  $h_{r,l}$  is the relative humidity in the top soil layer,  $c_{v,sat}$  is the saturated vapour concentration  $[m^3(H_2O) m^{-3}(air)]$ ,  $D_{v,l}$  is the vapour diffusivity in the top soil layer  $[m^2 s^{-1}]$ ;  $\phi_l$  is the

liquid matric flux potential  $[m^2 s^{-1}]$ ;  $K_I$  is the hydraulic conductivity of the top soil layer  $[m s^{-1}]$ ;  $\phi_{min}$  $[m^2 s^{-1}]$  is the matric flux potential corresponding to minimum soil moisture potential, set here to  $h_{min}$  $= -10^6$  m.  $E_{pol}$  comes from the solution of the coupled energy and moisture conservation equations at the soil-air interface with relative humidity at the surface set to 1 (Haverd and Cuntz, 2010; Haverd et al., 2013).

**Improved parameterization of in-canopy resistance to turbulent transfer**

We adapt the CABLE2.0 formulation of  $r_{soil}$  to account for local (in-canopy) stability effects on the resistance of transfer from the ground to the canopy air-space, effectively increasing the resistance 10 when ground sensible heat fluxes are negative. The adaptation splits the resistance into the sum of two components: the first  $r_{soil,a}$  from the soil roughness height to a shear height  $z_{sh}$ , and the second  $r_{soil,b}$  from  $z_{sh}$  to the displacement height d. We assume that the shear height, representing the depth of the shear-driven surface layer that forms along the ground surface under the canopy, is a small fraction of the canopy height, here 0.1. Both resistance components, like the original rsoib (Equation (18)) are integrals over the inverse of the Eddy diffusivity  $K_f$ :

5

$$r_{soil,a} = \int_{z_{0s}}^{z_{sh}} \frac{dz}{K_f(z)}$$
(30)

$$r_{soil,b} = \int_{z_{sh}}^{d} \frac{dz}{K_f(z)}$$
(31)

where alternate forms of the Eddy diffusivity are specified, the first accounting for local stability effects, and the second is the same as in the original formulation of  $r_{soil}$ .

$$20 K_{f}(z) = \begin{cases} \frac{\kappa z \widetilde{u}_{*}}{\Phi_{h}\left(\frac{z}{\widetilde{L}}\right)} & , z_{0s} < z < z_{sh} \\ \frac{1}{\sigma_{w}^{2} \tau_{L}} & , z_{sh} < z < d \end{cases}$$
(32)

This yields

$$r_{soil,a} = \widetilde{u_*} \int_{z_{0s}}^{z_{sh}} \frac{\Phi_h\left(\frac{Z}{\widetilde{L}}\right)}{\kappa z} dz$$
$$= \widetilde{u_*} \left[ \ln\left(\frac{z_{sh}}{z_{0s}}\right) - \psi_h\left(\frac{z_{sh}}{\widetilde{L}}\right) + \psi_h\left(\frac{z_{0s}}{\widetilde{L}}\right) \right]$$

(33)

and

$$r_{soil,b} = \frac{1}{u_*} \ln\left(\frac{d}{z_{sh}}\right) \frac{\exp\left(2c_{s,w}L\right)(d/h)}{a_3^2 c_{TL}}$$
(34)

In Equations (32)-(34),  $\kappa$  is the von Karman constant (0.4),  $\Phi_h$  is the Monin-Obukhov stability function (Garratt, 1992),  $\tilde{u_*}$  is the friction velocity at height  $z_{sh}$  and is related to the friction velocity at the reference height above the canopy by the same factor that attenuates the mean windspeed in the canopy:

5
$$\widetilde{u_*} = u_* \exp\left\{-c_u \left(1 - \frac{z_{sh}}{h}\right)\right\}$$
 (35)

where  $c_u$  is the exponent for an assumed exponential wind profile (Raupach, 1994). L is the local Obukhov length, correspondingly given by:

$$\tilde{L} = \frac{-u}{\kappa \frac{g}{T_{K}} \frac{H_{soil}}{\rho_{a} c_{p}}}$$
(36)

where g is the gravitational constant and  $T_K$  is the canopy air temperature [K].

**10 3 Data**

~ 3

Following the PLUMBER land surface model benchmarking experiment described by Best et al. (2015), we use data from 18 Eddy covariance flux tower sites, available as part of the FLUXNET LaThuile free fair-use subset (fluxdata.org; see Acknowledgements). Best et al. (2015) selected sites for broad coverage of vegetation types and climate, and we use the same sites here, with the

15 exception of five omissions (ElSaler and ElSaler2 (irrigated); Loobos (missing GPP observations), Palang (poor energy closure) and Merbleue (wetland site)), and three inclusions (Roccarespampani, Tharandt and Castelporzanio), such that our list of sites includes all 5 sites employed by De Kauwe et al. (2015) for their assessment of CABLE drought response during the 2003 European heatwave. Gap-filling and quality control were applied, as described by Best et al. (2015). Fluxes were 20 aggregated to monthly and daily values for comparison with model output.

FLUXNET site locations, IGBP plant functional type and data duration are listed in Table 2, combining information from Best et al. (2015) and De Kauwe et al. (2015).

| Name           | Country       | Lat            | Lon                | CABLE PFT            | Duration  |
|----------------|---------------|----------------|--------------------|----------------------|-----------|
| Amplero        | Italy         | 41.90 °N       | 13.61 °E           | C3 Grassland         | 2003-2006 |
| Blodgett       | United States | $38.90\ ^{o}N$ | 120.63 °W          | Evergreen Needleleaf | 2000-2006 |
| Bugac          | Hungary       | 46.69 °N       | 19.60 °E           | C3 Grassland         | 2002-2006 |
| Castelporziano | Italy         | 41.70 °N       | 12.37 °W           | Evergreen Broadleaf  | 2001-2006 |
| Espirra        | Portugal      | 38.64 °N       | $8.60$ $^{\rm o}W$ | Evergreen Broadleaf  | 2001-2006 |
| Fort Peck      | United States | 48.31 °N       | 105.10 °W          | C3 Grassland         | 2000-2006 |
| Harvard        | United States | 42.54 °N       | 72.17 °W           | Deciduous Broadleaf  | 1994-2001 |

|--|

| France        | 48.67 °N                                                                                                                                       | 7.06 °E                                                                                                                                                                                        | Deciduous Broadleaf                                                                                                                                                                                                                                                                                                                                                                                                                                                                                                                                   | 1999-2006                                                                                                                                                                                                                                                                                                                                                                                                                    |
|---------------|------------------------------------------------------------------------------------------------------------------------------------------------|------------------------------------------------------------------------------------------------------------------------------------------------------------------------------------------------|-------------------------------------------------------------------------------------------------------------------------------------------------------------------------------------------------------------------------------------------------------------------------------------------------------------------------------------------------------------------------------------------------------------------------------------------------------------------------------------------------------------------------------------------------------|------------------------------------------------------------------------------------------------------------------------------------------------------------------------------------------------------------------------------------------------------------------------------------------------------------------------------------------------------------------------------------------------------------------------------|
| Australia     | 12.49 °S                                                                                                                                       | 131.15 °E                                                                                                                                                                                      | C4 Grassland                                                                                                                                                                                                                                                                                                                                                                                                                                                                                                                                          | 2002-2005                                                                                                                                                                                                                                                                                                                                                                                                                    |
| United States | 45.20 °N                                                                                                                                       | $68.74\ ^{o}W$                                                                                                                                                                                 | Evergreen Needleleaf                                                                                                                                                                                                                                                                                                                                                                                                                                                                                                                                  | 1996-2004                                                                                                                                                                                                                                                                                                                                                                                                                    |
| Finland       | 61.85 °N                                                                                                                                       | 24.29 °E                                                                                                                                                                                       | Evergreen Needleleaf                                                                                                                                                                                                                                                                                                                                                                                                                                                                                                                                  | 2001-2004                                                                                                                                                                                                                                                                                                                                                                                                                    |
| South Africa  | 25.02 °S                                                                                                                                       | 31.50 °E                                                                                                                                                                                       | C4 grassland                                                                                                                                                                                                                                                                                                                                                                                                                                                                                                                                          | 2003-2004                                                                                                                                                                                                                                                                                                                                                                                                                    |
| Botswana      | 19.92 °S                                                                                                                                       | 23.56 °E                                                                                                                                                                                       | C4 Grassland                                                                                                                                                                                                                                                                                                                                                                                                                                                                                                                                          | 199-2001                                                                                                                                                                                                                                                                                                                                                                                                                     |
| Italy         | 42.40 °N                                                                                                                                       | 11.92 °W                                                                                                                                                                                       | Deciduous Broadleaf                                                                                                                                                                                                                                                                                                                                                                                                                                                                                                                                   | 2002-2006                                                                                                                                                                                                                                                                                                                                                                                                                    |
| United States | 46.24 °N                                                                                                                                       | $89.35\ ^{o}W$                                                                                                                                                                                 | Deciduous Broadleaf                                                                                                                                                                                                                                                                                                                                                                                                                                                                                                                                   | 2002-2005                                                                                                                                                                                                                                                                                                                                                                                                                    |
| Germany       | 58.97 °N                                                                                                                                       | $13.57 \ ^{\circ}W$                                                                                                                                                                            | Evergreen Needleleaf                                                                                                                                                                                                                                                                                                                                                                                                                                                                                                                                  | 1998-2005                                                                                                                                                                                                                                                                                                                                                                                                                    |
| Australia     | 38.66 °S                                                                                                                                       | 148.15 °E                                                                                                                                                                                      | Evergreen                                                                                                                                                                                                                                                                                                                                                                                                                                                                                                                                             | 2002-2005                                                                                                                                                                                                                                                                                                                                                                                                                    |
|               |                                                                                                                                                |                                                                                                                                                                                                | Broadleaf                                                                                                                                                                                                                                                                                                                                                                                                                                                                                                                                             |                                                                                                                                                                                                                                                                                                                                                                                                                              |
| United States | 48.56 °N                                                                                                                                       | $84.71\ ^{o}W$                                                                                                                                                                                 | Deciduous Broadleaf                                                                                                                                                                                                                                                                                                                                                                                                                                                                                                                                   | 1999-2003                                                                                                                                                                                                                                                                                                                                                                                                                    |
|               | France
Australia
United States
Finland
South Africa
Botswana
Italy
United States
Germany
Australia
United States | France48.67 °NAustralia12.49 °SUnited States45.20 °NFinland61.85 °NSouth Africa25.02 °SBotswana19.92 °SItaly42.40 °NUnited States46.24 °NGermany58.97 °NAustralia38.66 °SUnited States48.56 °N | France         48.67 °N         7.06 °E           Australia         12.49 °S         131.15 °E           United States         45.20 °N         68.74 °W           Finland         61.85 °N         24.29 °E           South Africa         25.02 °S         31.50 °E           Botswana         19.92 °S         23.56 °E           Italy         42.40 °N         11.92 °W           United States         46.24 °N         89.35 °W           Germany         58.97 °N         13.57 °E           United States         48.56 °S         148.15 °E | France48.67 °N7.06 °EDeciduous BroadleafAustralia12.49 °S131.15 °EC4 GrasslandUnited States45.20 °N68.74 °WEvergreen NeedleleafFinland61.85 °N24.29 °EEvergreen NeedleleafSouth Africa25.02 °S31.50 °EC4 grasslandBotswana19.92 °S23.56 °EC4 GrasslandItaly42.40 °N11.92 °WDeciduous BroadleafGermany58.97 °N13.57 °WEvergreen NeedleleafAustralia38.66 °S148.15 °EEvergreenUnited States48.56 °N84.71 °WDeciduous 
[revised manuscript text omitted]

10

15